# Synthesis, Characterization, and Optimization Studies of Starch/Chicken Gelatin Composites for Food-Packaging Applications

**DOI:** 10.3390/molecules27072264

**Published:** 2022-03-31

**Authors:** Jorge Iván Castro, Diana Paola Navia-Porras, Jaime Andrés Arbeláez Cortés, José Herminsul Mina Hernández, Carlos David Grande-Tovar

**Affiliations:** 1Grupo de Investigación SIMERQO, Departamento de Química, Universidad del Valle, Calle 13 No. 100-00, Santiago de Cali 76001, Colombia; jorge.castro@correounivalle.edu.co; 2Grupo de Investigación Biotecnología, Facultad de Ingeniería, Universidad de San Buenaventura Cali, Carrera 122 # 6-65, Santiago de Cali 76001, Colombia; dpnavia@usbcali.edu.co (D.P.N.-P.); valencia.mayra@correounivalle.edu.co (J.A.A.C.); 3Grupo de Materiales Compuestos, Escuela de Ingeniería de Materiales, Facultad de Ingeniería, Universidad del Valle, Calle 13 No. 100-00, Santiago de Cali 76001, Colombia; jose.mina@correounivalle.edu.co; 4Grupo de Investigación de Fotoquímica y Fotobiología, Facultad de Ciencias, Universidad del Atlántico, Carrera 30 Número 8-49, Puerto Colombia 081008, Colombia

**Keywords:** composites, food packaging, films, gelatin, starch

## Abstract

The indiscriminate use of plastic in food packaging contributes significantly to environmental pollution, promoting the search for more eco-friendly alternatives for the food industry. This work studied five formulations (T1–T5) of biodegradable cassava starch/gelatin films. The results showed the presence of the starch/gelatin functional groups by FT-IR spectroscopy. Differential scanning calorimetry (DSC) showed a thermal reinforcement after increasing the amount of gelatin in the formulations, which increased the crystallization temperature (Tc) from 190 °C for the starch-only film (T1) to 206 °C for the film with 50/50 starch/gelatin (T3). It also exhibited a homogeneous surface morphology, as evidenced by scanning electron microscopy (SEM). However, an excess of gelatin showed low compatibility with starch in the 25/75 starch/gelatin film (T4), evidenced by the low Tc definition and very rough and fractured surface morphology. Increasing gelatin ratio also significantly increased the strain (from 2.9 ± 0.5% for T1 to 285.1 ± 10.0% for T5) while decreasing the tensile strength (from 14.6 ± 0.5 MPa for T1 to 1.5 ± 0.3 MPa for T5). Water vapor permeability (*WVP*) increased, and water solubility (*WS*) also decreased with gelatin mass rising in the composites. On the other hand, opacity did not vary significantly due to the films’ cassava starch and gelatin ratio. Finally, optimizing the mechanical and water barrier properties resulted in a mass ratio of 53/47 cassava starch/gelatin as the most appropriate for their application in food packaging, indicating their usefulness in the food-packaging industry.

## 1. Introduction

The food industry is one of the most significant users of single-use plastic from petroleum, contributing to solid waste [1]. This plastic can even degrade into finer particles (micro- and nano-plastics) that can enter the food chain and affect different trophic levels, including humans [2]. By 2050, it is projected that more than 33 million tons of plastic will end up accumulating in various ecosystems, especially in the ocean [3]. Therefore, searching for more environmentally friendly alternatives is imperative to avoid using these polymers [4]. In this sense, several investigations have been carried out into the production of films containing biodegradable components from extracts of *Pseudomonas oleovorans* bacteria [5], chicken feathers [6], soy protein [7], chicken breast protein [8], sunflower protein [9], spinach flour [10], corn starch–chitosan blends [11], among others. However, many biodegradable films have disadvantages in mechanical properties and high production costs. Therefore, a viable option is to produce cassava starch films due to their low price, availability, and oxygen-barrier properties [12,13].

Cassava or tapioca (*Manihot esculenta* Crantz) is a low-cost tuber native to Thailand, and is used for starch production. Starch is a polymer composed mainly of amylose and amylopectin. Both components consist of chains of D-glucose residues with bonds α-(1,4) that are interconnected through α-(1,6)-glycosidic bonds forming branches in polymers [14]. In addition, amylose is responsible for the isotropic, odorless, tasteless, colorless, non-toxic, and biodegradable film-forming characteristics [15,16]. However, the mechanical properties of cassava starch films are insufficient for food packaging due to their brittle and weak behavior when subjected to tensile stress. In addition, films produced from polysaccharides are susceptible to moisture [17]. The above have promoted the synthesis of films, including plasticizing additives such as sorbitol, glycerol, and polyethylene glycol, to improve flexibility [18].

Gelatin is a polymer obtained from collagen and is used to manufacture bioplastics. Its miscibility with water allows for the production of different matrices in the food-packaging industry [19,20]. There are two types of gelatins, depending on the preparation methodology. Gelatin A is obtained by acid extraction, and gelatin B is obtained by alkaline extraction [21]. The primary sources of gelatin are pig skins, pig bones, and cows’ skin [22]. Gelatin’s triple-helix protein structure and amino acid composition provide physical barrier capacity and protection against ultraviolet light, protecting foods from physical damage and oxidation caused by reactive oxygen species (ROS) [23,24,25].

The amino acid content in gelatin (mainly proline, hydroxyproline, and glycine) makes chicken gelatin a potential component for developing new materials applicable to the food industry. Chicken gelatin contains considerably higher gel strength in Bloom value (355 ± 1.48 g) than bovine gelatin (229 ± 0.71 g). However, there are few studies related to films incorporating cassava starch with gelatin in an optimized formulation to maximize the composite’s properties [26,27].

Veiga-Santos et al. found that the addition of gelatin did not significantly affect the mechanical properties of tapioca starch sheets plasticized with sucrose [27]. However, the addition of gelatin increased the water vapor permeability of the glycerol-plasticized cassava starch film [26]. There are no studies where optimized films from chicken gelatin and cassava starch ratio are formulated. Despite the reported applications of cassava starch bioplastics with chicken-based gelatin, there is a need to optimize the cassava starch–gelatin mixtures to maximize the packaging properties of the films obtained. In this study, five (T1–T5) formulations of cassava starch and chicken-waste gelatin-based films were developed and characterized in a simplex-lattice mixture design, optimized to determine the starch/gelatin mixture with desirable functional properties in food packaging. Therefore, this study aimed to obtain bioplastic films from mixtures of cassava starch and chicken gelatin, characterize them, and find an optimal blend of the components to generate a bioplastic film whose properties are helpful in food packaging.

## 2. Results and Discussion

### 2.1. FT-IR Spectroscopy Analysis of the Cassava Starch/Gelatin Films

Figure 1 shows the FT-IR characterization of the cassava starch/gelatin films. All samples showed higher intensity bands in the amide region for the films with high gelatin content (3287 a 3308 cm^−1^) due to the -OH groups from starch, gelatin, and water-adsorbed molecules [28].

The characteristic bands of cassava starch are present in T1. The symmetrical stress band corresponding to the -C-H group is observed around 2928–2930 cm^−1^. Additionally, stress bands associated with amide groups are overlapped by -OH stress bands. Bands at 3308 cm^−1^ and 2930 cm^−1^ for O-H and -C-H stretching vibrations are observed. At lower vibration frequencies, the bands associated with in-plane bending vibration for the CH_2_ group and bending of the C-OH group are also observed between 1413 and 1337 cm^−1^. Finally, the band corresponding to the stretching of the antisymmetric bridge of the C-O-C group rose at 1149 cm^−1^ [29].

Strong hydrogen bonds between starch and gelatin might generate smooth and homogeneous films, especially in the T3 formulation (50/50 starch/gelatin), as discussed in the morphological section. On the other hand, shifting at higher vibrational frequencies is observed for T2–T4 (starch/gelatin) films, especially for the strain band at 3295 cm^−1^ associated with the -OH group. In addition, asymmetric stretching bands of the amide I group were observed between 1637–1652 cm^−1^, increasing intensity with higher gelatin content [30]. Additionally, due to the presence of amino groups from the gelatin, the band at 3200 cm^−1^ is shifted. Additionally, gelatin’s -C=O group (1743–1746 cm^−1^) increased, and the CH-O-CH_2_ groups (1009–1033 cm^−1^) decreased as the gelatin concentration within the polymeric matrix increased (formulations T2–T4). The shifting of the -OH band and the increase in the amide I band is probably due to hydrogen bonding of the C=O and N-H groups with the -OH group, which demonstrates strong bonding between starch and gelatin components. The exact position of the amide I band is determined by the most stable backbone conformation and hydrogen-bonding pattern associated with the stretching vibrations of the C=O (70–85%) and C-N (10–20%) group [31]. On the other hand, formulation T5 showed the main band at 1742 cm^−1^, corresponding to the C=O group of amides-I. Likewise, the band located at 1544 cm^−1^ is attributed to amides-II rising from the stretching vibrations of the N-H groups and the stretching vibrations of the C-N group. Finally, the bands located at 3287 and 2928 cm^−1^ correspond to the stretching vibrations of the N-H group for an amide-I and the symmetric tensions of the aliphatic carbons CH_2_, respectively [32].

### 2.2. Thermal Analysis of Cassava Starch/Gelatin Films

Thermogravimetric analysis (TGA) aims to evaluate the degradation of any material with increasing temperature. Figure 2 shows the degradation profiles of the different films. The cassava starch/chicken gelatin films (T1, T2, T4, and T5) presented three stages of degradation, while T3 presented four stages of degradation. The first stage of degradation corresponds to weight loss between 25 and 150 °C, attributed to the loss of absorbed bound water in the films and low-molecular-weight materials [33]. Additionally, this degradation step is attributed to the reaction between the carbonyl groups of starch and the amino groups of gelatins [34]. This behavior indicates that thermoplastic processing is not recommendable for starch and gelatin films due to side reactions.

The second stage occurs between 150 and 270 °C, corresponding to the decomposition of organic materials of low molecular weight attributed to cassava starch [33,35]. In this sense, it is highlighted that the gelatin-containing films exhibited lower values in this degradation stage compared to T1 (decreasing from 317 to 311 °C), which did not contain gelatin. According to Figure 2, it is likely that crosslinking was generated during the first heating stage through intramolecular reactions as the temperature increased. Therefore, films containing a higher amount of gelatin facilitated the formation of a higher degree of crosslinking, as observed in the DTGA in Figure 2 (undefined degradations) [34]. Likewise, a continuous thermal degradation was observed for the films containing a higher concentration of starch (T2 and T3), reflected in the decrease in the degradation temperatures compared to T1 [34]. Finally, the third stage of degradation occurs between 270 and 600 °C, attributed to the total decomposition of the material (polysaccharides, protein, and glycerol volatilization), including some contained gases (CO_2_, CO, H_2_O) and carbon compounds present in the material [11]. On the other hand, formulation T3 shows a small shoulder of the fourth degradation stage centered at 388 °C. This may correspond to the elimination of hydrogen functional groups, degradation, and depolymerization of the starch–gelatin interactions between the polymer chains, which can be formed due to the higher gelatin amount present in the film that at higher temperatures causes crosslinking and strong interactions with starch, requiring more energy for bond-breaking [36].

The low-resolution degradations observed in the DTGA of samples T4 and T5 are due to crosslinking reactions between carbonyl and amino groups, degrading at low speed and high temperatures. Consequently, the processing of these sheets has limitations related to possible uncontrolled responses at high temperatures [34,37]. On the other hand, the thermal analysis of the films containing a higher amount of gelatin shows that they are susceptible when the temperature exceeds 100 °C, as indicated by the thermogravimetric derivative analysis (DTGA, blue line, Figure 2).

Differential scanning calorimetry (DSC) studies the phase transitions of the material as the temperature increases [38]. DSC analysis of cassava starch/chicken gelatin films (Figure 3) exhibited two endothermic peaks associated with the melting of starch crystalline regions during retrogradation [39]. The two endothermic peaks are attributed to each film’s two melting temperatures (Tm). For T1, the Tm was observed at 190 °C. However, the Tm increased to 208 °C and 206 °C for T2 and T3. The increase in the Tm might be due to the above-discussed crosslinking reactions between the C=O and N-H functional groups from starch and gelatin, reducing the mobility of the polymer chains and generating thermostable films [40].

On the other hand, higher amounts of gelatin introduce a higher degree of flexibility, which affects the observation of the well-defined endothermic melting peak of the material [41]. For T4, the two peaks are poorly defined due to the higher amount of gelatin (75%) compared to starch (25%) in the formulation. These observations are consistent with previous studies using chicken gelatin as a polymeric matrix [40,42]. Additionally, formulation T5 shows a decrease in structural stability, because its melting peaks (157 and 198 °C) are at a lower temperature with an approximate difference of 8 °C compared to formulations T2 and T3. Therefore, the collagen structure in chicken gelatin involves breaking hydrogen bonds and helix coil transition between polypeptide chains adjacent to collagen molecules in the denaturation process [43].

### 2.3. Scanning Electron Microscopy (SEM) of Cassava Starch/Gelatin Films

Figure 4 shows the scanning electron micrographs for the cassava starch/gelatin films. A homogeneous, continuous, and smooth surface appearance (characteristic of polysaccharide films such as starch) was observed for T1 (starch film) due to the excellent molecular packing. For the T5 gelatin film, a rough, heterogeneous, and discontinuous appearance is observed, because low (<4 wt.%)- or high (>16 wt.%)-protein films present an inadequate film formation. A low concentration of proteins generates lower macromolecule interactions, while high-globular-protein concentration reduces the water activity, creating films with greater elongations and tear resistance [44].

In formulations T2–T4, rough surfaces with pores could be observed. The presence of starch reduced the heterogeneous and irregular appearance of the gelatin film (T5). However, with the introduction of gelatin, a roughness and heterogeneous surface are observed because of the spherical protein presence that intercalates between the polysaccharide chains, forming a rounded appearance. When the gelatin content increases, more intense molecular interaction between starch and gelatin occurs, decreasing the rounded appearance with a heterogeneous aspect.

Formulation T3 (50/50 starch/gelatin) showed the slightest rough appearance, demonstrating good compatibility with this formulation. However, there were differences in the appearance of the starch/gelatin films. Apparent differences are likely due to hydrogen bonds between the components [11,37]. These differences in the microstructure can be associated with the different organization of the polymeric chains during composite formation [40]. Similarly, the roughness of cassava starch/bovine gelatin type B films was observed when the gelatin concentration was increased [45].

### 2.4. The Tensile Strength of the Cassava Starch/Gelatin Films

Table 1 shows the values of Young’s modulus, tensile strength, and deformation percentage of the cassava starch/gelatin films. Significant differences (*p* < 0.05) in the characterization of the mechanical properties between the samples are evident. These results indicate that the sample’s flexibility significantly increased with the increase in the amount of gelatin, decreasing, in turn, the stiffness of the material, as observed in the Young’s modulus values. The stiffness of the films, shown in Table 1, significantly reduced with increasing gelatin (*p* < 0.05) between T2, T3, and T4 formulations. Young’s modulus for the films ranged from 2.2 ± 1.1 MPa for T4 to 440.4 ± 46.1 MPa for T2, a behavior previously observed [40,46], attributed to the formation of new hydrogen bonds between the functional groups of cassava starch and gelatin, which in turn decrease the number of hydrogen bonds between starch chains. During film drying, the formation of intra- and inter-molecular hydrogen bonds and chemical crosslinking is facilitated by the supply of thermal energy, and increases with temperature. As the polymeric network encounters a higher amount of starch, the stiffness of the films also significantly increases [17,45]. Despite that, a higher proportion of starch needs a more elevated amount of glycerol in the formulation, which increases the film’s flexibility. The addition of gelatin in the matrix promotes the formation of new hydrogen, increasing the material’s ductility.

On the other hand, tensile strength decreased significantly (*p* < 0.05) as gelatin concentration increased, from 6.2 ± 0.1 MPa for T2 (cassava starch/gelatin 75/25 wt.%) to 1.4 ± 0.2 MPa for T4 (cassava starch/gelatin 25/75 wt.%). These results agree with the significantly increased (*p* < 0.05) flexibility, from 4.6 ± 0.3% for T2 to 157.8 ± 16.2% for T4. From these observations, it is evident that the increase in the amount of gelatin in the composite films significantly decreases the stiffness of the composite. The presence of gelatin proteins with globular characteristics distorts the homogeneous molecular arrangement of the starch chains [40].

### 2.5. Water Vapor Permeability (WVP) of the Cassava Starch/Gelatin Films

The water vapor permeability (*WVP*) values of cassava starch/gelatin films are described in Table 2. Statistically, there were significant differences (*p* < 0.05) between the treatments evaluated. The treatment without cassava starch, containing only gelatin (T5), presented the lowest value, while the treatment with 100% cassava starch (T1) showed the highest value. It can be inferred that the variation in gelatin/starch concentration in the films affects water vapor permeability. The increase in cassava starch concentration significantly (*p* < 0.05) increased *WVP* values. A higher proportion of starch increases the active binding points with water molecules, favoring water vapor transport through the films. In addition, the higher the ratio of starch, the higher the proportion of plasticizer, which attracts more water molecules due to its polar character and its contribution to the formation of hydrogen bonds. These results are consistent with recent studies reporting an increase in *WVP* values with increasing cassava starch proportion in films obtained from chicken gelatin and cassava starch [40], and fish gelatin-based films, where the increase in the proportion of rice flour increased the *WVP* values [21]. On the other hand, treatments T2, T3, and T4, which had starch/gelatin ratios of 75/25, 50/50, and 25/75, respectively, were significantly different. This behavior could be related to the interactions formed between gelatin and starch that occupy both polymers’ active sites, affecting the passage of water vapor through the crosslinked network.

Food packaging has different functions to prevent or reduce the moisture gain of the packaged food inside, so low *WVP* values are expected. Table 2 shows that the *WVP* of the films ranged from 0.365 to 0.437 g mm kPa^−1^ h^−1^ m^−2^. These values are similar to those reported for films made with cassava starch and gelatin between 0.16 and 0.318 g mm kPa^−1^ h^−1^ m^−2^ [47] and lower than those written by other authors in chicken gelatin films with cassava starch of 1.9 to 5.8 g mm kPa^−1^ h^−1^ m^−2^ [40], and bovine gelatin and cassava starch films of 2.7 a 6.4 g mm kPa^−1^ h^−1^ m^−2^ [48]. However, conventional materials with a low water vapor barrier include polyamide 0.027 to 0.03 g mm kPa^−1^ h^−1^ m^−2^ and low-density polyethylene 0.003 to 0.0035 g mm kPa^−1^ h^−1^ m^−2^. The film’s values obtained in this study are higher, indicating a high water vapor permeability.

### 2.6. Water Solubility of the Cassava Starch/Gelatin Films

The water solubility values of the cassava starch/gelatin films are shown in Table 2. A significant effect of the cassava starch-gelatin ratio on the water solubility of the films was evident (*p* < 0.05). An increasing trend in solubility was observed with increasing gelatin concentration in the films. The treatment without gelatin (T1) presented a significantly low value, while 100% gelatin film (T5) showed a significant increase. However, treatment T4 (75/25 gelatin/cassava starch) showed the highest solubility value, indicating that the rise in gelatin concentration increases the interactions with water, possibly due to an increase in the formation of hydrogen bridges that favor the bonding between the bioplastic and the water molecules. Likewise, the plasticizer increases with increased starch proportion, incrementing interactions with water given its polar character. However, these interactions are higher when gelatin increases in the formulation, which is associated with the creation of new hydrogen bonds.

The water solubility results of the cassava starch/gelatin films ranged from 3.67 to 5.15% for treatments T1 to T3, similar to those reported in a study of cassava starch and fish gelatin-based films [49]. Treatments T4 and T5 showed values between 13.45 and 21.26%, like those found in sheets made with corn starch and gelatin [47] and in films made with corn starch, gelatin, and glycerol or sorbitol [50]. The behavior of the water solubility of the cassava starch/gelatin films may be related to the fact that the increase in the amount of gelatin in the mixture increases the gelation and compaction rates of the polymeric matrix, which decreases the solubility, as evidenced in the T3 treatment. Treatments T1 to T3 with starch/gelatin proportions 100/0, 75/25, and 50/50 differed significantly from T4 with 25/75. This difference may be because the hydroxyl group (O-H) and the amide group (-C=O-N-H) of gelatin enable the formation of strong hydrogen bonds with the O-H groups of starch [45]. According to Zhang et al. [51], the hydrogen bonds reduce the ability of gelatin to interact with water molecules, decreasing solubility. However, when the proportion of gelatin is unbalanced concerning the proportion of starch, solubility increases, probably because the excess polymer promotes solubility since it has the possibility of forming more bonds with water through available non-cross-linked polar groups.

### 2.7. Opacity of the Cassava Starch/Gelatin Films

The opacity values of the cassava starch/gelatin films are shown in Table 2. The variation in starch/gelatin concentration did not significantly (*p* > 0.05) affect the opacity values of the cassava starch/gelatin films. This indicates that the interactions between the main suspension components, cassava starch and gelatin, within the polymer matrix did not significantly affect the penetration of light through the films. The opacity values for the five treatments were between 21.20 and 22.50%. There have been previous reports with similar values for potato starch and gelatin films [52], corn starch and gelatin [53], and cassava starch and gelatin [47].

Sensory characteristics play a fundamental role in the selection of packaging materials. Opacity is a property that, in addition to impacting appearance, influences the shelf life of the packaged product. For some packaged foods, films with good opacity provide high lightfastness and therefore help to improve the shelf life of photosensitive foods. However, high levels of opacity are not always required. According to the opacity values found, the films were slightly opaque due to aromatic amino acids in the gelatin acting as a barrier for light [21] and the crystallinity of the starch molecules [52].

### 2.8. Optimization of the Cassava Starch/Gelatin Films

The mechanical property values of a conventional low-density polyethylene film [54] were used as a target to optimize the properties of the cassava/gelatin starch films in this study. Considering the application in food packaging, low water vapor permeability and water solubility values are required since the film will directly contact food. Consequently, *WVP* and water solubility (*WS*) values were minimized. Table 3 shows the maximum and minimum values used in optimizing each response variable (y). For each variable, a goal was established.

The optimized predictions for each response variable are shown in Figure 5. The optimization shows a prediction percentage that is considered acceptable (91.44%). The predicted values for the variables *WVP*, *WS*, Young’s modulus, tensile strength, and deformation were 0.4 g mm/kPa h m^2^, 8.98%, 203.7 MPa, 3.4 MPa, and 28.4%, respectively, with predictability between 0.64 < d < 0.96. The above is obtained in films with a cassava starch/gelatin proportion of 53/47.

## 3. Materials and Methods

### 3.1. Materials

Cassava starch (*Manihot esculenta* Crantz) was purchased from Tecnas S.A. (Cali, Colombia). Sodium hydroxide and glacial acetic acid were purchased from Merck (Burlington, MA, USA). Extraction of gelatin from chickens (from chicken feet supplied by a local farmer) was reported elsewhere [39]. Food-grade glycerol was purchased from Merck (Burlington, MA, USA). All chemicals were analytical, and no further purification was performed unless otherwise indicated.

### 3.2. Methods

#### 3.2.1. Preparation of the Sample of Chicken Gelatin

Chicken gelatin was prepared according to the methodology previously reported [39], with slight modifications. The chicken feet were washed, disinfected, and immersed in 400 mL of a 0.25 M NaOH solution for six hours. Subsequently, the sample was transferred to a 4% (*v*/*v*) glacial acetic acid solution for three hours. Then, a heat treatment was conducted at 80 °C for eight hours in a ratio of 60 g of chicken feet in 40 mL of water. Finally, it was filtered, and the resulting solid was dehydrated in an oven at 55 °C until the material was obtained.

#### 3.2.2. Preparation of Cassava Starch/Gelatin Films

Composite films’ preparation followed Podshivalov’s methodology [55]. The cassava starch suspension was initially heated at 85 °C with constant stirring for one hour. Later, the cassava starch was mixed with 25% *w*/*w* glycerol (concerning the weight of cassava starch) to room temperature for 10 min. Subsequently, the suspension was heated to 80 °C for 15 min, cooled to 60 °C, and then we slowly added the gelatin. Finally, the rest of the suspension was sonicated at 60 °C for 50 min using an ultrasonic bath (Branson, Madrid, Spain). All the blends and the simplex-lattice design are reported in Table 4.

The suspensions in plastic molds were environmentally cured for 24 h and placed for another 24 h in an oven at 40 ± 0.2 °C. After that, the solid film of cassava starch/gelatin was pilled off. The samples were placed in an airtight glass container at 50% relative humidity (RH) until the test time. The samples were cut for mechanical testing according to ASTM D6287, ASTM D618, and ASTM D882. All the thicknesses were measured in a digital micrometer. Film thicknesses ranged from 1.16 to 1.33 mm (±0.17).

#### 3.2.3. Film Characterization

##### Thermal Analysis

Thermal gravimetric analysis was run on a TA Instruments TGA Q50 V20.13 Build 39 (TA instrument, New Castle, DE, USA) between 30 and 700 ± 2 °C using film samples (5~10 mg) under nitrogen at a flow rate of 60 mL/min of and a heating rate of 10 °C/min. The fusion (Tm) temperatures were determined using a DSC2A-00181 system (TA Instruments) with a heating/cooling rate of 5 °C/min from −25 °C to 250 °C and cooling again to −25 °C using the differential scanning calorimetry (DSC) technique. TGA and DSC results were analyzed using the TA Instruments’ Universal Analysis software.

##### Functional Group Characterization of the Films

The functional group of the films was studied using FTIR spectroscopy on an IR Affinity-1 spectrometer (Shimadzu, Kyoto, Japan). The spectra were obtained using a 4 cm^−1^ resolution and 32 scans in the range of 500 to 4000 cm^−1^. A diamond tip and ATR (attenuated total reflectance) mode were used for the test.

##### Morphology Analysis

The surface morphologies were scanned with an electron microscope (JSM-6490LA, JEOL) at 20 kV acceleration voltage, using a copper coating with the mode of secondary backscattered electrons. A gold layer was used for electron conduction on the samples.

##### Mechanical Properties

A universal SHIMADZU EZ-LZ test machine (Shimadzu, Tokyo, Japan) was used for the tensile test, following the ASTM D882 standard. A 500 N load cell was used. The samples were tested with a gap of 100 mm between jaws at a speed of 50 mm/min. At least 10 samples per formulation were used. The length and width of the tested samples were 100 mm and 20 mm, respectively.

##### Water Vapor Permeability

Water vapor permeability (*WVP*) determination followed the ASTM-E96. The diameter of the tested samples was 80 mm. The samples were glued to the mouth of a permeation cell containing silica gel. The permeation cells were stored inside the desiccator with 73 ± 2% RH at 25 °C. The slope of the permeation cell mass change as a function of time was obtained by linear regression. *WVP* (g mm/kPa h m^2^) was calculated as follows in Equation (1):(1)$$WVP=\left[\frac{WVTR}{PR\times H}\right]\times l$$
where *WVTR* is the water vapor transmission rate obtained from the ratio between the water gained as a function of time (g/h) and the permeation area of the sample (m^2^). *P* is the saturation vapor pressure of water (kPa); *RH* is the relative humidity in the desiccator, and *l* is the average sample thickness (m).

##### Solubility in Water

Solubility in water (*WS)* was determined by following the method described by Cheng et al. [56]. Dried films (105 °C for 24 h) of 2 cm × 2 cm were weighted (*Wi*). Then they were hydrated with 50 mL of distilled water at room temperature for 24 h with random agitation. Finally, samples were filtered and dried at 105 °C to obtain their final dry weight (*Wf*). *WS* (%) was calculated as follows in Equation (2):(2)$$WS=\frac{Wi-Wf}{Wi}\times 100$$

##### Opacity

The opacity test was developed using a colorimeter (Konica Minolta, Japan). Data were processed by Spectra Magic NX software. Film samples with 2 cm × 10 cm were placed under aperture for opacity measurement, determined by using black (*P_b_*) and white patron (*P_w_*) as references. Opacity (*Op*) was calculated as the ratio between the opacity (*P_b_*) and opacity (*P_w_*). Both *P_b_* and *P_w_* were determined in five films with black and white standards (%) using Equation (3).
(3)$$Op\left(\%\right)=\frac{P_{b}}{P_{w}}\times 100$$

#### 3.2.4. Experimental Design and Optimization

A simplex-lattice design was used with two components and four lattices (*m*). The design points are given by *m* + 1. The five design points correspond to:

*x_i_* = 0, 1/*m*, 2/*m*… *m*/*m*, where *i* is the number of components, leaving the following mixtures:

(*x*_1_, *x*_2_): (0, 1); (0.25, 0.75); (0.5, 0.5); (0.75, 0.25); (1, 0).

The mixtures are shown in Table 4.

The optimization was performed by setting the mechanical properties of a commercial polyethylene film as a target and minimizing the values of the *WVP*, and *WS* found in the blends. The desirability function (D: global; d: individual) that converts the functions to a scale between 0 and 1 was used, combining them using the geometric mean and optimizing the general metric means.

The experimental design and the optimization were performed using Minitab (2019) software.

##### Statistical Analysis

We used an analysis of variance (ANOVA) to establish statistical differences between treatments. Fisher’s test was used as a post-ANOVA analysis to compare the means between treatments. ANOVA and post-ANOVA were obtained using Minitab (2019) software.

## 4. Conclusions

Cassava starch/gelatin composite films were obtained, with good mechanical properties comparable with reported values of commercial polyethylene films. The analysis of the FT-IR spectra for the composites showed the presence of hydroxyl and amide groups characteristic of starch and gelatin and with shifts of their bands at various ratios due to hydrogen bonds. Thermogravimetric analysis of the films showed that with a high amount of gelatin (T4 and T5), degradation profiles were poorly defined, characteristic of films with high flexibility. In the DSC analysis, it was evidenced that the endothermic peak Tc increased due to the presence of gelatin. Still, for T4 and T5, there was a notable decrease in its value, probably due to molecular disorder between the chains and greater flexibility.

Additionally, the morphological analysis of T2 and T3 showed minor roughness, discontinuity, and heterogeneity, and a decrease in the flexibility of gelatin compared to the heterogeneous appearance of the gelatin film (T5). The results of material stiffness through Young’s modulus suggest that the flexibility of the films increased with the increase in the amount of gelatin, decreasing, in turn, the stiffness of the material (from 2.9 ± 0.5 to 285.1 ± 10.0 MPa for formulations T1 and T5, respectively). On the other hand, the water solubility results of the cassava starch/gelatin films were between 3.67 and 21.26% for treatments T1 and T5, respectively, similar to those found for cassava starch- or cornstarch-based films with gelatin.

With all these characterization results for the composites, an optimal formulation was obtained to develop cassava starch/gelatin-based films in a 53/47 ratio, plasticized with glycerol using the casting method, that would meet the expectations of the model polyethylene film for food-packaging applications. Young’s modulus, tensile strength, deformation, thermal, *WVP*, and water solubility variables were affected by the cassava starch and gelatin mixtures evaluated in the treatments. Based on the predicted values for each response in the optimization, it can be inferred that these films could be used as food-packaging material, as their mechanical properties are close to those of low-density polyethylene. Future studies could incorporate other additives to improve moisture stability properties such as *WVP*.

## Figures and Tables

**Figure 1 molecules-27-02264-f001:**
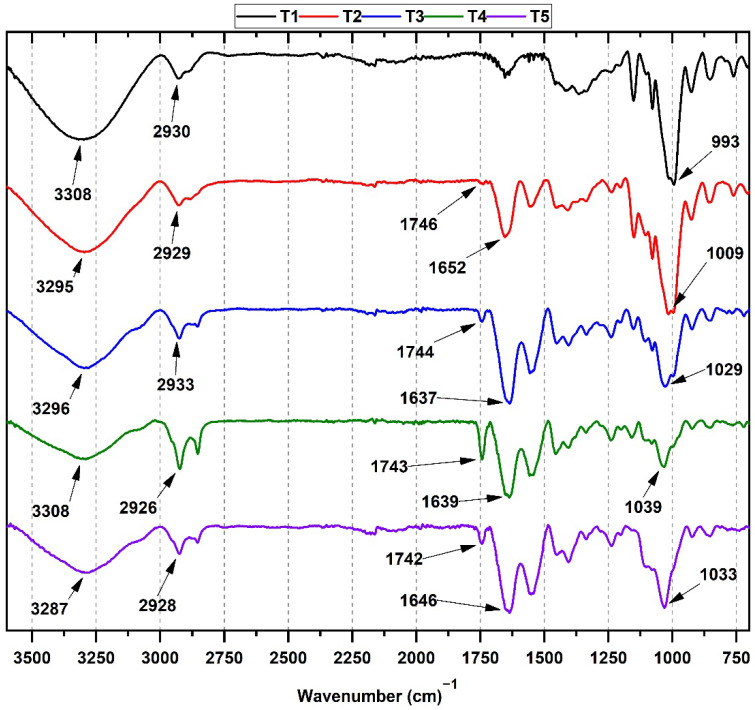
FTIR spectra of cassava starch/gelatin films. T1 gelatin/cassava starch 0/100; T2 gelatin/cassava starch 25/75; T3 gelatin/cassava starch 50/50; T4 gelatin/cassava starch 75/25; T5, gelatin/cassava starch 100/0.

**Figure 2 molecules-27-02264-f002:**
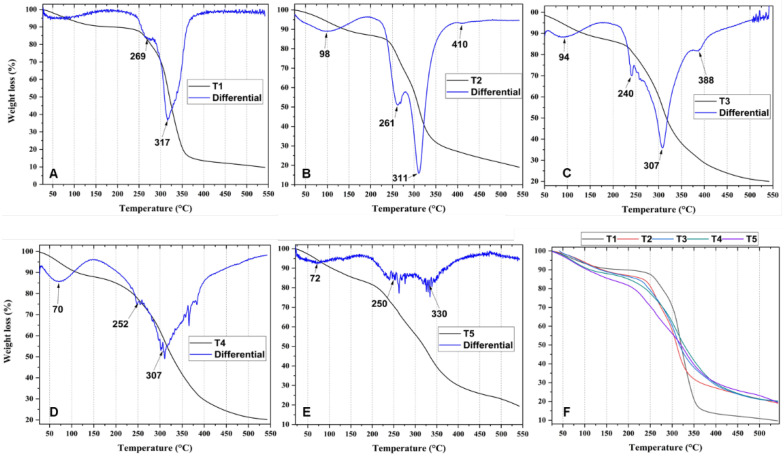
Thermogravimetric analysis for T1, (**A**) gelatin/cassava starch 0/100; T2, (**B**) gelatin/cassava starch 25/75; T3, (**C**) gelatin/cassava starch 50/50; T4, (**D**) gelatin/cassava starch 75/25; T5, (**E**) gelatin/cassava starch 100/0; (**F**) All the samples.

**Figure 3 molecules-27-02264-f003:**
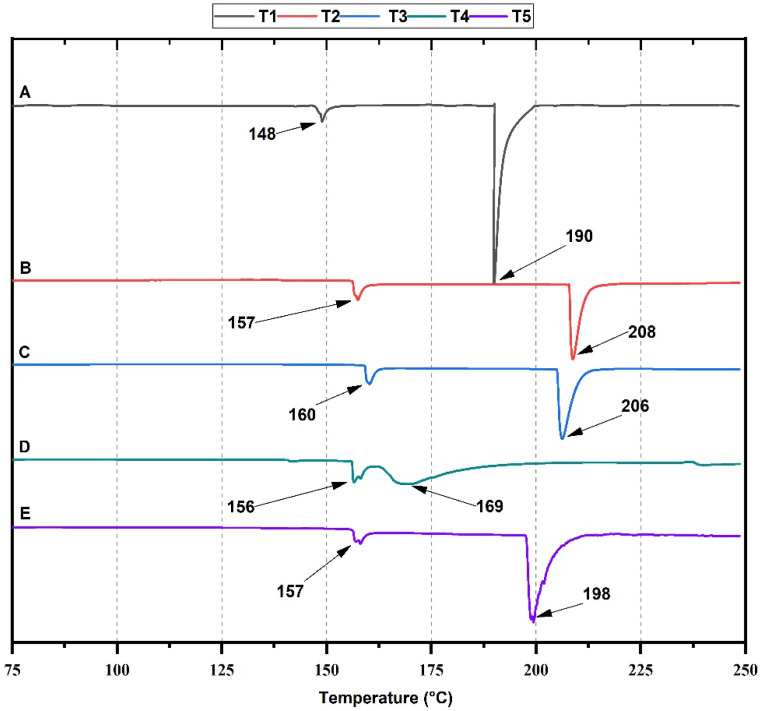
DSC thermogram of films: T1, (**A**) gelatin/cassava starch 0:100; T2, (**B**) gelatin/cassava starch 25/75; T3, (**C**) gelatin/cassava starch 50/50; T4, (**D**) gelatin/cassava starch 75:25; T5, (**E**) gelatin/cassava starch 100/0.

**Figure 4 molecules-27-02264-f004:**
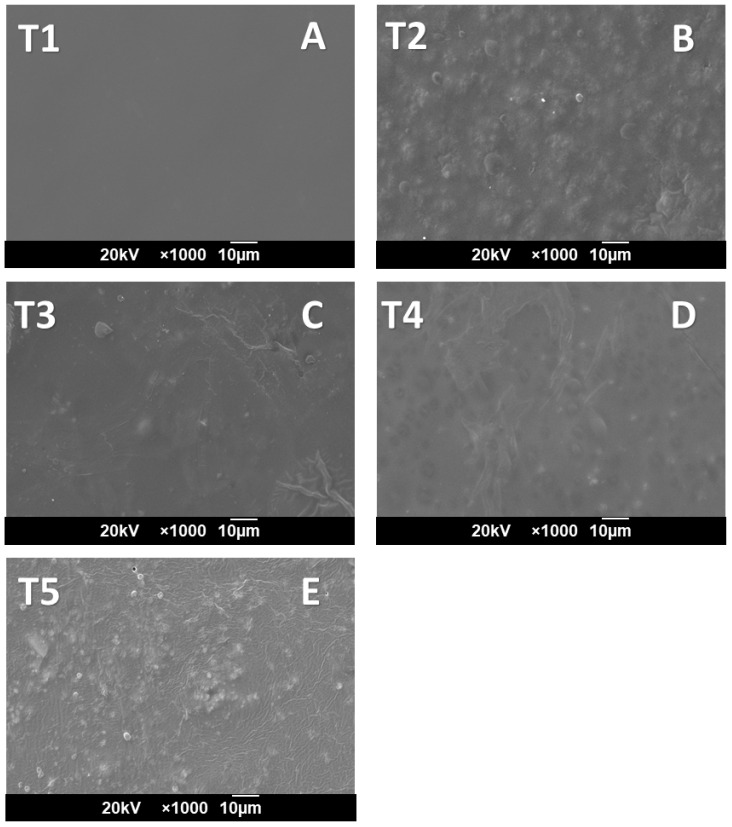
Morphological analysis of slides by scanning electron microscopy (SEM): T1, (**A**) gelatin/cassava starch 0:100; T2, (**B**) gelatin/cassava starch 25/75; T3, (**C**) gelatin/cassava starch 50/50; T4, (**D**) gelatin/cassava starch 75/25; T5, (**E**) gelatin/cassava starch 100/0.

**Figure 5 molecules-27-02264-f005:**
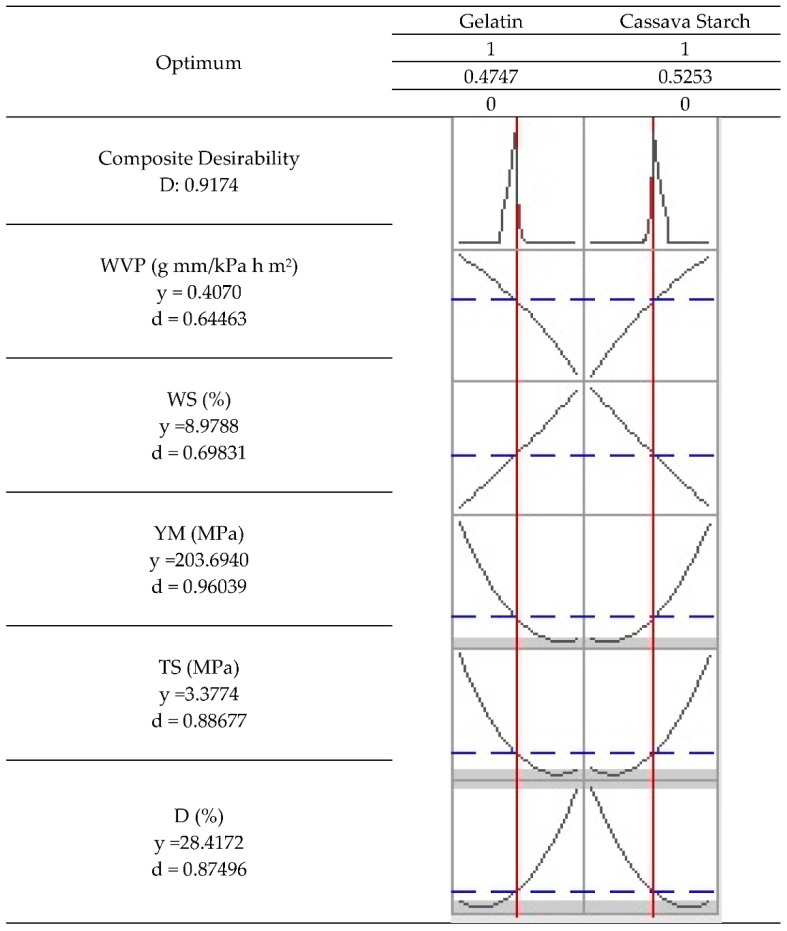
Simultaneous optimization of the variables in the cassava starch/gelatin films.

**Table 1 molecules-27-02264-t001:** Mechanical properties of cassava starch/gelatin films: T1 (gelatin/cassava starch 0/100; T2 gelatin/cassava starch 25/75; T3 gelatin/cassava starch 50/50; T4 gelatin/cassava starch 75/25; T5 gelatin/cassava starch 100/0.

Formulation	Young’s Modulus (MPa) *	Tensile Strength (MPa) *	Deformation (%) *
T1	1116.0 ^a^ ± 57.6	14.6 ^a^ ± 0.5	2.9 ^a^ ± 0.5
T2	440.4 ^b^ ± 46.1	6.2 ^b^ ± 0.1	4.6 ^a^ ± 0.3
T3	246.5 ^c^ ± 21.9	3.7 ^c^ ± 0.3	9.0 ^a^ ± 0.8
T4	2.2 ^d^ ± 1.1	1.4 ^d^ ± 0.2	157.8 ^b^ ± 16.2
T5	0.4 ^d^ ± 0.0	1.5 ^d^ ± 0.3	285.1 ^c^ ± 10.0

* Different letters in the same column indicate significant differences (*p* < 0.05).

**Table 2 molecules-27-02264-t002:** Water vapor permeability, water solubility, and opacity properties of cassava starch/gelatin films.

Formulation	*WVP* (g mm/ kPa h m^2^) *	Water Solubility (%) *	Opacity (%) *
T1	0.437 ^a^ ± 0.11	4.20 ^a^ ± 2.12	21.20 ^a^ ± 0.62
T2	0.395 ^b^ ± 0.03	5.15 ^a^ ± 1.29	22.11 ^a^ ± 0.82
T3	0.409 ^b^ ± 0.04	3.67 ^a^ ± 1.31	21.21 ^a^ ± 0.92
T4	0.408 ^b^ ± 0.04	21.26 ^b^ ± 0.97	22.5 ^a^ ± 0.84
T5	0.365 ^c^ ± 0.03	13.45 ^c^ ± 1.61	22.37 ^a^ ± 0.98

* Different letters in the same column indicate significant differences (*p* < 0.05).

**Table 3 molecules-27-02264-t003:** Optimization conditions.

Response Variable	Minimum	Maximum	Goal
*WVP* (g mm/ kPa h m^2^)	0.365	0.437	Minimum
*WS* (%)	3.67	21.26	Minimum
Young’s modulus (MPa)	0.4	1116	200
Tensile strength (MPa)	1.5	14.68	8
Deformation (%)	2.9	285.1	100

**Table 4 molecules-27-02264-t004:** Simplex-lattice experimental design for the formation of the cassava starch/gelatin composite films.

Sample	Gelatin (wt.%)	Cassava Starch (wt.%)
T1	0	100
T2	25	75
T3	50	50
T4	75	25
T5	100	0

## Data Availability

Data are available under request to the corresponding author.

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
