# Peer review of "Synthesis, Characterization, and Optimization Studies of Starch/Chicken Gelatin Composites for Food-Packaging Applications"

_molecules, 2022, doi:10.3390/molecules27072264_

Round 1

Reviewer 1 Report

The work is well written, the results are clearly presented. The concept of optimizing the parameters of the produced films, i.e. an attempt to match these parameters to low-density polyethylene films, seems interesting because the end user is usually not interested in the details of film production, but in the compatibility of its functional properties.

It is a pity that the authors chose SEM for morphological analysis. Certainly, the use of AFM would bring richer information, the more so as the image for T1 in Figure 4 is only a gray field.

In chapters 3.2.2 and 3.2.3, the authors cite ASTM standards, however, in the scientific work, the reader directly expects information about the thickness of the film, its width, etc. rather than just ensuring that the measurements conform to the standards. This information should be given in the work.

A minor correction in line 94 is needed - 100/0 instead of 0/100.

Author Response

Dear Editor,

We want to submit our corrected version of the paper entitled "Synthesis, Characterization, and Optimization Studies of Starch/Chicken Gelatin Composites for Food Packaging Applications." The paper was authored by Jorge Iván Castro, Diana Paola Navia-Porras, Jaime Andrés Arbeláez Cortés, José Herminsul Mina Hernandez, and Carlos David Grande-Tovar, who agreed with all the corrections. The corrections are presented below point by point in red for easy comprehension.

Reviewer 1

Comments and Suggestions for Authors

  1. The work is well written. The results are clearly presented. The concept of optimizing the parameters of the produced films, i.e., an attempt to match these parameters to low-density polyethylene films, seems interesting because the end-user is usually not interested in the details of film production but in the compatibility of its functional properties.

R// We appreciate the positive comments from the reviewer about the goal of our work.

  1. It is a pity that the authors chose SEM for morphological analysis. Certainly, the use of AFM would bring richer information, the more so as the image for T1 in Figure 4 is only a gray field.

R// We appreciate the suggestion from the reviewer. However, we agree that the AFM technique might bring in-depth information or even cross-section analysis by SEM, especially for the cross-section film. However, the image of T1 in Figure 4 was obtained according to the described methodology for SEM acquisition. The surface is smooth and homogeneous as compared to the other films. In our case, we wanted to demonstrate surface morphologies differences because of the new polymer interactions. Even though the surface differences are clear enough to support our point, we believe that the reader can clearly distinguish between the morphologies.

  1. In chapters 3.2.2 and 3.2.3, the authors cite ASTM standards; however, in the scientific work, the reader directly expects information about the thickness of the film, its width, etc., rather than just ensuring that the measurements conform to the standards. This information should be given in the work.

R// We appreciate the valuable suggestion from the reviewer. The correction was applied in lines 402, 439, 440, and 445.

  1. A minor correction in line 94 is needed - 100/0 instead of 0/100.

R// We appreciate the suggestion from the reviewer. The correction was applied.

Reviewer 2 Report

The present manuscript deals with the preparation and characterization of composite films based on cassava starch gelatin mixtures. The manuscript has some weakness.

Firstly, it is not clear which is the aim of the work and which are the advances carried out in the present work in relation to starch-gelatin films in comparison to the cited works from Veiga-Santos et al and Tongdeesoontorn et al. The authors should better define and clarify the aim and the meaning of the work in the introduction, especially in relation to the proposed application in food packaging. The optimization of the formulation conditions carried out is also poor. Particularly, it is not clear how the desiderable functional properties and their values were selected to carried out the “centroid experimental design”.

The description of several methods is not accurate and incomplete.

No information regarding chemical properties of the starch and gelatin used.

Extraction method for gelatin is not clear. Which is 60% (w/v) acqueous solution?

Method for the experimental design and the software used is not reported

“Solubility in water (WS)” seems a not appropriated terminology, maybe “water adsorption”

The method for opacity is not clear

In conclusions section, the prepaed films are claimed to have good compatibility and mechanical properties comparable to typical polyethylene films. However, compatibility was not investigated and polyethylene films were not analysed.

Captions of Figure 3 and Figure 4 report the referring letters (A, B, C, D, E), but these letters do not appear in the figures.

Line 181 How the discontinuous appearance of the gelatin film is related to its flexibility?

Line 305-306 Are the authors sure that the opacity is not related to a not optimal compatibility among starch and gelatin?

Author Response

Dear Editor,

We want to submit our corrected version of the paper entitled "Synthesis, Characterization, and Optimization Studies of Starch/Chicken Gelatin Composites for Food Packaging Applications." The paper was authored by Jorge Iván Castro, Diana Paola Navia-Porras, Jaime Andrés Arbeláez Cortés, José Herminsul Mina Hernandez, and Carlos David Grande-Tovar, who agreed with all the corrections. The corrections are presented below point by point in red for easy comprehension.

Reviewer 2

The present manuscript deals with the preparation and characterization of composite films based on cassava starch gelatin mixtures. The manuscript has some weaknesses.

  1. Firstly, it is not clear which is the aim of the work, and which are the advances carried out in the present work in relation to starch-gelatin films in comparison to the cited works from Veiga-Santos et al. and Tongdeesoontorn et al. The authors should better define and clarify the aim and the meaning of the work in the introduction, especially in relation to the proposed application in food packaging.

R// We appreciate the suggestion from the reviewer. The correction was applied in the introduction on lines 79, 80, and 86-89.

  1. The optimization of the formulation conditions carried out is also poor. Mainly, it is not clear how the desired functional properties and their values were selected to carry out the "centroid experimental design."

R// We appreciate the suggestion from the reviewer. The targets of the optimization and its methodology were clarified in lines 484 to 490. A description of the experimental design was also given. It was initially mentioned as a centroid mixture design because the mixtures have a center, but it was adjusted as a simplex-lattice design and described between lines 472 to 490.

  1. The description of several methods is not accurate and incomplete.

R// We appreciate the suggestion from the reviewer. The methodological description of the Thermal (412-418), FTIR (425-426), Opacity (lines 463-469), and Mechanical  (lines 439 to 440) tests was adjusted.

  1. No information regarding the chemical properties of the starch and gelatin used.

R// We appreciate the suggestion from the reviewer. Since our objective was to prepare films, we directly prepared them using cassava starch and gelatin, as described in section 3.2.2. We focused on the film's characterization since we propose them for food-packaging applications. You can find the ATR-FTIR spectra in Figure 1 from the functional groups of starch  (100%) films (T1) and gelating (100%) films (T5), as well as the discussion of the chemical composition between lines 100-107 and lines 122-127.

  1. The extraction method for gelatin is not clear. Which is 60% (w/v) aqueous solution?

R// We appreciate the suggestion from the reviewer. The gelatin extraction methodology was adjusted in line 383: "…Then, a heat treatment was conducted at 80 °C for eight hours in a ratio of 60 g of chicken feet in 40 mL of water. Finally, it was filtered, and the resulting solid was de-hydrated in an oven at 55 °C until the material was obtained."

  1. Method for the experimental design and the software used is not reported

R// We appreciate the suggestion from the reviewer. The experimental design adjusted the technique, including the software used (Lines 472 to 490).

  1. "Solubility in water (WS)" seems a not appropriated terminology, maybe "water adsorption."

R// We appreciate the suggestion from the reviewer. However, we kindly want to refer that we disagree with this statement. The reason is that the term "water adsorption" is commonly used when water adsorption occurs on the surface in a temperature-dependent manner (adsorption isotherms). Water adsorbed molecules can interact with the materials' surface through different interactions (physical, electrostatic, non-covalent forces) and with a monolayer or multi-layer fashion. Water adsorption is usually measured by gravimetry after placing the film in environments with relative humidity at a fixed temperature to obtain an adsorption isotherm.

On the other hand, water solubility measures the amount of film solubilized in water when the film is immersed in water for 24 hours, as described in the methodology of this study. In other words, water adsorption deals with water molecules interacting on the film surface. In contrast, solubility deals with the water molecules passing through the surface and penetrating the bonds of the polymer matrix, surrounding and separating them from the polymer matrix, causing a decrease in the film's weight.

Here are some studies where both concepts are discussed:

  • Merci et al., 2019. Films based on cassava starch reinforced with soybean hulls or microcrystalline cellulose from soybean hulls. Food Packaging and Shelf life
  • Gómez-Aldapa et al., 2020. Effect of Polyvinyl alcohol on the physicochemical properties of biodegradable starch films. Materials Chemistry and Physics
  • Nandi and Guha, 2021. Organic acid-compatibilized potato starch/guar gum blend films. Materials Chemistry and Physics

  1. The method for opacity is not clear

R// We appreciate the suggestion from the reviewer. The methodological description of the Opacity test was adjusted (lines 462 to 470).

  1. In the conclusions section, the prepared films are claimed to have good compatibility and mechanical properties comparable to typical polyethylene films. However, compatibility was not investigated, and polyethylene films were not analyzed.

R// We appreciate the suggestion from the reviewer. Although polyethylene was not analyzed, it was taken as a target sample based on reported mechanical properties since it is a material commonly used in food packaging. The values of the mechanical properties of low-density polyethylene were referenced in section 2.8. Lines 497-498 were corrected accordingly.

  1. Captions of Figure 3 and Figure 4 report the referring letters (A, B, C, D, E), but these letters do not appear in the figures.

R// We appreciate the suggestion from the reviewer. The letters were added to the figures.

  1. Line 181: How is the gelatin film's discontinuous appearance related to its flexibility?

R// We appreciate the suggestion from the reviewer. Within the text, the corresponding correction can be explicitly found on lines 190-195 and lines 207-212:

"For the T5 gelatin film, a rough, heterogeneous and discontinuous appearance is observed because protein films below 4% and above 16% present inadequate films, which can be explained by the fact that a lower concentration of protein coexists with a lower molecular interaction. Otherwise, a higher protein concentration reduces the water activity due to a high concentration of macromolecules. The foregoing translates into films with greater elongations and tear resistance".

 "However, with the gelatin introduction, a roughness and heterogeneous surface is observed because of the rounded protein presence, that intercalates between the polysaccharide chains, forming a spherical appearance. When the gelatin content increases, more intense molecular interaction between starch and gelatin occurs, decreasing the spherical appearance but with a heterogeneous aspect. "

  1. Line 305-306: Are the authors sure that the opacity is not related to not optimal compatibility between starch and gelatin?

R// We appreciate the suggestion from the reviewer. We consider it impossible to relate the opacity result to a potential lack of compatibility between components since there were no significant differences between the treatments, as was the case with the other tests. For this reason, we only refer to the presence of aromatic groups of amino acids and starch crystals that affects transparency, probably as UV scavengers or light diffraction from polymer aggregates.

Reviewer 3 Report

The development and creation of biodegradable polymeric materials is one of the most promising areas and a reasonable alternative to plastic polymers, so the article Synthesis, characterization, and optimization studies of starch/chicken gelatin composites for food packaging applications may be of interest to readers of Molecules. This is a good publication, carefully compiled, well written and well explained, with a lot of work done, contains useful data and is recommended for publication after minor comments are corrected.

 Results and discussion

Can the resolution of Figure 2 be improved?

Films (T1, T2, T4 and T5) represent three stages of degradation, while T3 represents four stages of degradation. What happens in the fourth stage of degradation if complete decomposition of the material is already observed in the third stage?

If the presence of starch improves film heterogeneity, then why does sample T2 with 75% starch appear rougher and more heterogeneous than sample T3 and T4?

Besides gel strength, are there other benefits to using chicken gelatin?

Does the film thickness increase with increasing gelatin concentration?

What effect does a plasticizer have on the physicochemical properties of films?

Author Response

Dear Editor,

We want to submit our corrected version of the paper entitled "Synthesis, Characterization, and Optimization Studies of Starch/Chicken Gelatin Composites for Food Packaging Applications." The paper was authored by Jorge Iván Castro, Diana Paola Navia-Porras, Jaime Andrés Arbeláez Cortés, José Herminsul Mina Hernandez, and Carlos David Grande-Tovar, who agreed with all the corrections. The corrections are presented below point by point in red for easy comprehension.

Reviewer 3

The development and creation of biodegradable polymeric materials are one of the most promising areas and a reasonable alternative to plastic polymers, so the article "Synthesis, characterization, and optimization studies of starch/chicken gelatin composites for food packaging applications" may be of interest to readers of Molecules. This is a good publication, carefully compiled, well written, and well explained, with a lot of work done, contains valuable data, and is recommended for publication after minor comments are corrected.

R// We appreciate the positive comments from the reviewer of our work.

  1. Can the resolution of Figure 2 be improved?

R// We appreciate the suggestion from the reviewer. The resolution of the image was corrected, lines 141-143.

  1. Films (T1, T2, T4, and T5) represent three stages of degradation, while T3 represents four stages of degradation. What happens in the fourth stage of degradation if the complete decomposition of the material is already observed in the third stage?

R// We appreciate the suggestion from the reviewer. We added more discussion between lines 158-163.

  1. If the presence of starch improves film heterogeneity, then why does sample T2 with 75% starch appear rougher and more heterogeneous than samples T3 and T4?

R// We appreciate the suggestion from the reviewer. Surface analysis of starch films (T1) shows flat and homogeneous surfaces. We added the following information between lines 207-211: "However, with the gelatin introduction, a roughness and heterogeneous surface is observed because of the spherical protein presence, that intercalates between the polysaccharide chains, forming a rounded appearance. When the gelatin content increases, more intense molecular interaction between starch and gelatin occurs, decreasing the globular appearance but with a heterogeneous aspect. "

  1. Besides gel strength, are there other benefits to using chicken gelatin?

R// We appreciate the suggestion from the reviewer. Gelatin extracted from chicken feet has benefits in addition to gel strength in film formation. These benefits are easy incorporation during the elaboration process of the film-forming suspension as it does not require aggressive agitation processes or other additional methods. On the other hand, the gelatin was obtained from low-quality chicken feet considered waste and can be used for this type of bioplastics production.

  1. Does the film thickness increase with increasing gelatin concentration?

R// We appreciate the suggestion from the reviewer. The thickness data of the films were incorporated in section 3.2.2. There was no evidence of an increase in the thickness of the films due to the incorporation of gelatin. The last result was associated with the ease of formation of the film-forming suspension in all treatments. 

  1. What effect does a plasticizer have on the physicochemical properties of films?

R// We appreciate the suggestion from the reviewer. The effect of the plasticizer was incorporated in sections  2.4 (lines 241 to 245), 2.5 (lines 271 to 273), and 2.6 (lines 307 to 310).

Round 2

Reviewer 2 Report

The authors have addressed the reviewer comments and the manuscript is now suitable for publication